

# DES-YOLO: a novel model for real-time detection of casting surface defects

Chengjun Wang[1,*], Jiaqi Hu[2,3,*], Chaoyu Yang[1] and Peng Hu[1]

[1] School of Artificial Intelligence, Anhui University of Science and Technology, Huainan, Anhui, China
[2] School of Computer Science and Engineering, Anhui University of Science and Technology, Huainan, Anhui, China
[3] School of Information Science and Engineering, Wuhan University of Science and Technology, Wuhan, Hubei, China
[*] These authors contributed equally to this work.

## ABSTRACT

Surface defect inspection methods have proven effective in addressing casting quality control tasks. However, traditional inspection methods often struggle to achieve high-precision detection of surface defects in castings with similar characteristics and minor scales. The study introduces DES-YOLO, a novel real-time method for detecting castings' surface defects. In the DES-YOLO model, we incorporate the DSC-Darknet backbone network and global attention mechanism (GAM) module to enhance the identification of defect target features. These additions are essential for overcoming the challenge posed by the high similarity among defect characteristics, such as shrinkage holes and slag holes, which can result in decreased detection accuracy. An enhanced pyramid pooling module is also introduced to improve feature representation for small defective parts through multi-layer pooling. We integrate Slim-Neck and SIoU bounding box regression loss functions for real-time detection in actual production scenarios. These functions reduce memory overhead and enable real-time detection of surface defects in castings. Experimental findings demonstrate that the DES-YOLO model achieves a mean average precision (mAP) of 92.6% on the CSD-DET dataset and a single-image inference speed of 3.9 milliseconds. The proposed method proves capable of swiftly and accurately accomplishing real-time detection of surface defects in castings.

## INTRODUCTION

Casting is a critical method for manufacturing various heavy-duty and precision components for mechanical systems. Although there are strict process quality controls and inspections in the core process links, such as molding, pouring, and shakeout, it is difficult to fully automate the casting manufacturing process due to real-time monitoring and detection technology limitations. During the casting process, various surface defects may manifest (*Nadot, Mendez & Ranganathan, 2004*; *Chen et al., 2021*), including fractures and voids resulting from suboptimal casting practices, anomalies associated with inadequate temperature control, and abrasions from transportation and handling. The surface

Corresponding author
Chengjun Wang,
2001055@aust.edu.cn

defects substantially compromise the functionality and longevity of the cast products. Consequently, detecting and identifying surface flaws in cast materials are vital to reducing manufacturing costs and enhancing the quality of the final products.

Methods for detecting metal anomalies include artificial visual inspection detection (*Zhang et al., 2020*), non-destructive testing (*Strecker, 1983*; *Silva et al., 2023*), and image processing techniques (*Pastor-López et al., 2021*; *Xing & Jia, 2021*; *Wang, Li & W, 2022*; *Nieniewski, 2020*; *Li et al., 2023b*). Image processing substantially reduces labor costs, minimizing the inconsistencies and influence of subjective evaluations. However, existing studies primarily focus on metal surface defects (*Liu, Zhang & Dong, 2023*), with limited research focusing on applying deep learning to casting surface defect detection. Within the domain of defect identification, *Xiao, Wu & Hu (2020)* proposed a two-stage defect detection method based on the Mask R-CNN network, utilizing the IPCNN model. However, this approach faces real-time challenges in capturing and detecting small target defect features. *Cheng et al. (2022)* introduced the DS-Cascade R-CNN model, incorporating a spatial attention mechanism and deformable convolution at multiple scales. Despite these innovations, the two-stage algorithm's inherent limitations hinder detection speed.

Single-stage detection frameworks, exemplified by the YOLO series (*Redmon & Farhadi, 2018*; *Wang, Yeh & Liao, 2021*; *Long et al., 2020*; *Wang, Bochkovskiy & Liao, 2023*), employ convolutional neural networks (CNNs) to simultaneously estimate the coordinates of target bounding boxes and classification probabilities. These methodologies significantly enhance detection speed and reduce the dependency on extensive hardware resources, making it well-suited for real-time surface defect detection. *Li et al. (2023a)* employed techniques like an enhanced channel attention mechanism and multi-spatial pyramid pooling for automatic defect identification in wire arc additive manufacturing. *Parlak & Emel (2023)* utilized the YOLO model approach to study internal defects in high-pressure aluminum die-casting. Their research showcased effective defect identification through an end-to-end learning process. *Xing & Jia (2021)* utilized a convolutional network classification model (SCN) with a symmetric module and three convolutional branches, achieving impressive average accuracy rates of 99.61% and 95.84% in identifying surface defects in raw aluminum castings. However, limited dataset availability impacted generalization ability. *Xie et al. (2023)* investigated the FE-YOLO construct, integrating depth-wise separable convolution alongside an enhanced feature pyramid network for multi-scale object detection, achieving accuracy enhancements at the expense of escalated model complexity and protracted convergence. The RDD-YOLO model (*Zhao et al., 2023*), combining the Res2Net block and dual feature pyramid networks, addresses these challenges by decoupling the head for regression and classification tasks. *Yuan et al. (2023)* developed a method to enhance low-light images using CLAHE, markedly improving image contrast while avoiding shifts in chromaticity, thereby significantly enhancing the visibility of details under dim lighting. *Tang, Yu & Wu (2023)* introduced the RCID-YOLOv5s defect detection method, incorporating a specialized small target detection layer. This innovation addresses the challenge of low detection accuracy for minor defects and enhances the overall accuracy in identifying defects on railway catenary insulators.

Computer vision application for casting surface defect detection encounters distinct challenges, diverging from other industrial defect detection methods. Casting surface defects blend with the background, exhibit irregular shapes, vary significantly in type and size, and complicate feature extraction. Additionally, as network depth increases, features of more minor defects, such as pores, can become obscured, leading to detection difficulties. The scarcity of dedicated datasets for casting surface defects further constrains the diversity of defect types available for study, limiting research progress in this area.

A novel method has been developed to address the challenge of detecting surface defects in castings, with its effectiveness validated through the establishment of a well-balanced casting surface defect dataset (CSD-DET):

1. The introduction of the DSC-Darknet feature extraction network features a lightweight DSC layer and DBC-Block to construct the backbone network. Additionally, the GAM attention module is integrated to enhance feature discernment, improve extraction efficiency, and reduce computational demands.

2. We integrate the enhanced pyramid pool module to address the issue of losing features for small target samples, effectively expanding the feature receptive field.

3. A fusion of slim-neck architecture, combining GSConv and VoV-GSCSP, is utilized to refine the DES-YOLO model for lightweight and expedited real-time detection of casting surface anomalies.

4. The SIoU bounding box regression loss function is adopted, incorporating an area ratio penalty and vector angle to bolster model robustness and accelerate convergence in the optimization phase.

## MATERIALS & METHODS

This section introduces the critical components of the proposed DES-YOLO model, designed for the swift detection of casting surface defects. These components include the DSC-Darknet backbone, an enhanced pyramid pool module (EPPM), and a slim-neck architecture, as depicted in Fig. 1.

### DSC-Darknet backbone network

The CSP-Darknet53 backbone network is designed to leverage its residual structure to move convolutional layers across the original image, facilitating the fusion of various fine-grained image features. Our investigations have indicated that applying CSP-Darknet53 to detecting defects on casting surfaces does not markedly elevate detection precision; instead, it results in heightened computational load and extended training durations. We propose the DSC-Darknet feature extraction network to improve the accuracy of casting surface defect inspection models.

In the DSC-Darknet backbone network, traditional convolution layers are replaced with DSC convolutional layers, comprising Distributed Shift Convolution (DSConv (*Gennari, Fawcett & Prisacariu, 2019*)), batch normalization (BN), and the CELU activation function. DSConv, functioning as a plug-and-play convolution, employs a variable quantization kernel (VQK) and dual distribution shifts to substitute standard convolution layers in CNNs seamlessly. The VQK is engineered to store integer values of variable bit-length,

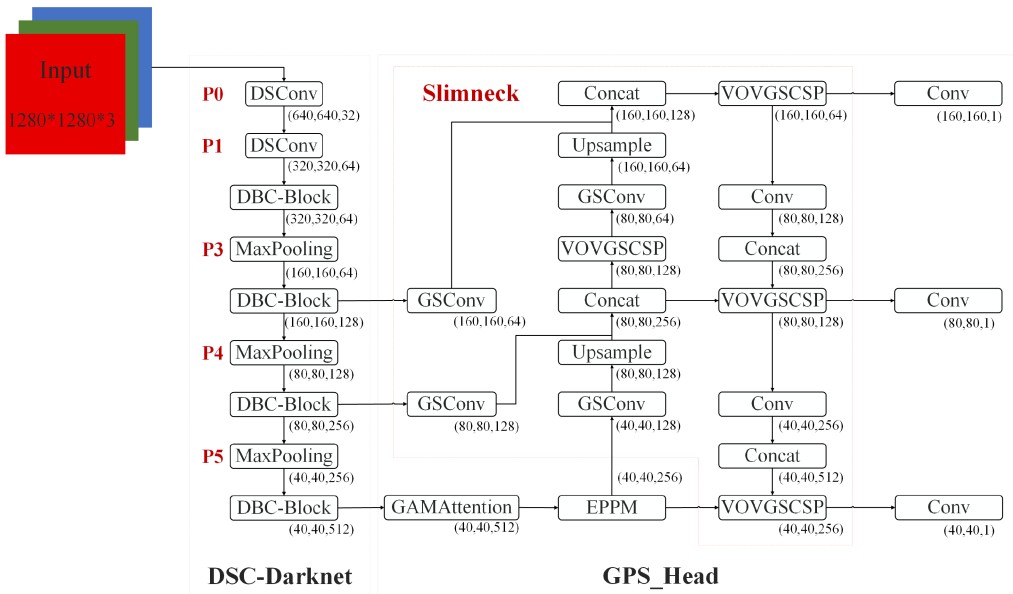

**Figure 1** DES-YOLO structure diagram.

thus minimizing floating-point calculations, reducing error accumulation, and lowering computational overhead. The distribution shift mechanism is segmented into the kernel shifter (KDS) and the channel distribution shifter (CDS). KDS adjusts the depth value of the VQK for each segment, while CDS consolidates channel values per segment. This structure enables the kernel-based and channel-based distribution shifters to replicate the outputs of the original convolution, which accelerates network convergence and enhances the model's generalization capabilities.

The DBC-Block design, illustrated in Fig. 2B, incorporates DConv and partitions the feature map into two divided into two independent branches. The primary pathway transmits features directly to the Conv layer, while the auxiliary path aggregates high-frequency features through the Conv and DSConv layers. Two independent branches implement feature aggregation at different scales using the Conv layer. This structure of the DBC-Block effectively addresses potential issues of gradient vanishing associated with increased network depth and simplifies the overall complexity of the DSC-Darknet backbone network.

DSC-Darknet generates feature information across five scales *via* a cascading feature transfer. As a high feature layer, P5 features rich semantic and contextual information but will lose the characteristic information of small defective targets. To address this, a global attention mechanism (GAM attention; *Liu, 2021*) is introduced following the P5 layer. This mechanism is designed to refine the detection of small targets by reducing background noise and evenly distributing feature data within the spatial domain. The GAM attention mechanism consists of two submodules: channel attention and spatial attention, illustrated in Fig. 3. The channel attention component employs a multilayer perceptron (MLP) to redistribute feature information and underscore interdimensional relationships. The

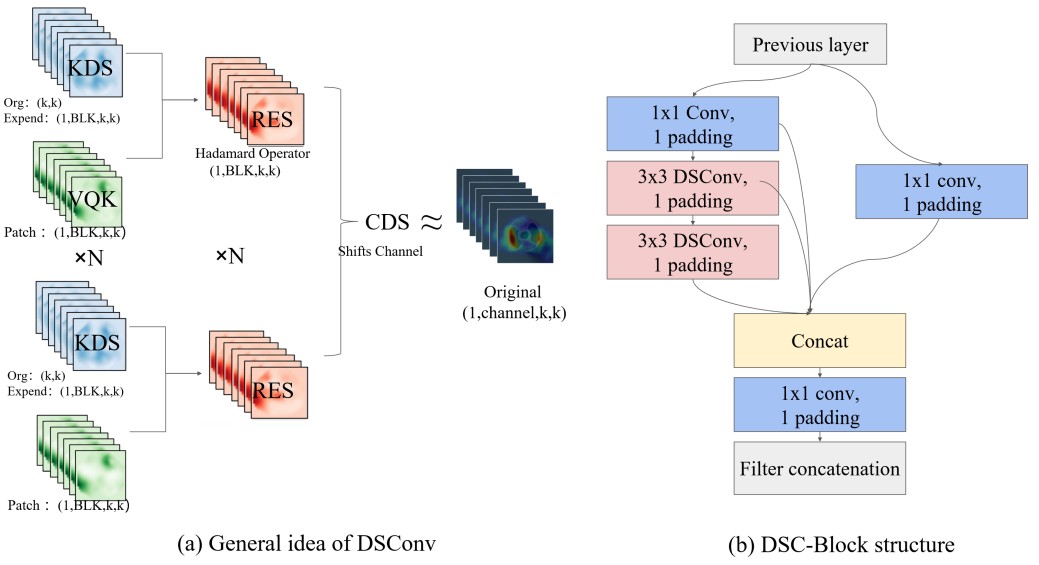

(a) General idea of DSConv

(b) DSC-Block structure

**Figure 2** **The primary components comprising DSC-Darknet.** (A) The VQK and two distributed shifts that ultimately render the result equivalent to the initial tensor, thus forming DSConv. (B) The architecture for DBC-Block aggregation.

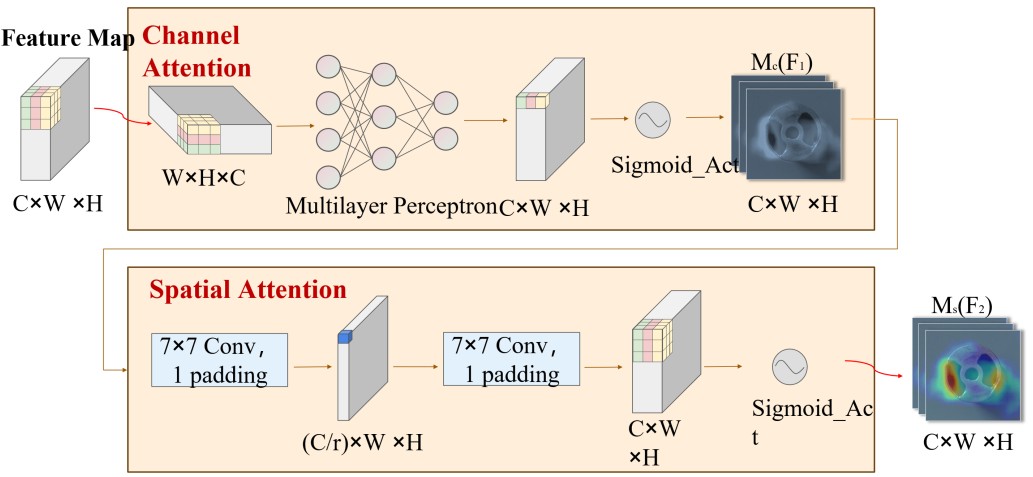

**Figure 3** **The GAM attention mechanism includes two submodules: channel attention and spatial attention.**

spatial attention submodule employs two convolutional layers aligned with channel data to enhance spatial details and focus on more minor target features.

## Enhanced pyramid pool module

Utilizing multiscale feature extracts within the DSC-Darknet framework may lead to reduced contextual information and the loss of fine-grained features across various subregions. The enhanced pyramid pooling module (EPPM) is integrated into the last

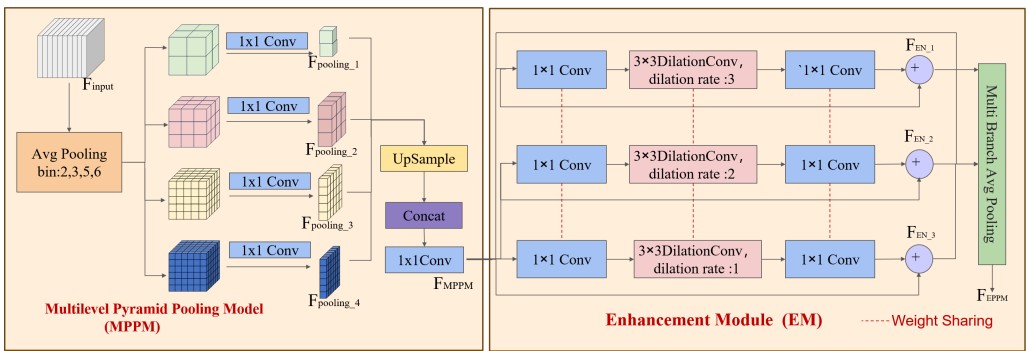

**Figure 4   EPPM structure.** The multilevel pyramid pooling model (MPPM) and the enhancement module (EM).

layer feature map to address this challenge. The EPPM is composed of the multilevel pyramid pooling model (MPPM) (*Zhao et al., 2016*) and the enhancement module (EM) (*Yu et al., 2022*). The model polymeric prior global contextual information enhances the flexibility of the feature extraction network and improves its ability to detect small targets.

The MPPM module addresses the loss of spatial information attributable to average pooling by using (2 × 2, 3 × 3, 5 × 5, 6 × 6) four-dimensional pooling branch to segment the feature map into distinct subregions, as depicted in Fig. 4A. To preserve global contextual details, the MPPM conducts convolution operations that reduce the contextual dimensions to a quarter of their original size following multilevel pooling. Each branch feature value is processed through the ReLU activation function to enhance the model's nonlinearity. The MPPM module utilizes bilinear up-sampling to return low-level features to their original dimensions to aggregate features without adding computational overhead. Ultimately, The MPPM module utilizes a cascading technique to integrate global contextual information with the multidimensional features of the local context, thereby enhancing the generation of robust global features, as detailed in Eq. (1). The input feature $F_{input}$ generates four pooled features $F_{pooling}$ with different dimensions after average pooling and 1 × 1 convolution operations with different bin values are set. The MPPM module integrates multiscale features by up-sampling and concatenation operations on branches, and achieves channel dimensionality reduction by 1 × 1 convolution operations to generate multiscale features $F_{MMPM}$.

$$F_{pooling\_i} = C((P(F_{input})))$$
$$F_{MPPM} = C(U(F_{pooling\_1}) + U(F_{pooling\_2}) + U(F_{pooling\_3}) + U(F_{pooling\_4})) \tag{1}$$

$U(X)$ represents a bilinear interpolation up-sampling operation applied to the feature map. $C(X)$ denotes a convolution operation with a kernel size of 1 and a step size of 1, while $P(X)$ stands for the average pooling of the feature map. The utilization of the average pooling preserves finer details in the image information, effectively addressing the challenge of increased variance in estimation values resulting from restricted neighborhood sizes.

The EM module incorporates convolution branches that feature varying expansion rates, residual links, and stratified weighting to expand the feature receptive field and address challenges such as reducing detail in small target features, gradient instability, and signal attenuation during the training phase. The multi-scale features of the MPPM part are passed through a convolutional layer with a kernel size of 1 to reduce the number of feature channels by half.

The multi-expansion rate convolution branch within the EM is designed to augment the semantic expressiveness of the receptive field without altering its size. This branch is structured with a three-branch structure, utilizing $3 \times 3$ convolutions and dilation convolutions with expansion coefficients of 1, 2, and 3. This arrangement conserves computing resources and diminishes the risk of overfitting through a weight-sharing strategy among the branches. Subsequently, the channel dimension of the feature map is enriched by a $1 \times 1$ convolutional layer, which ensures the restoration of the enhanced feature channels to the original image's dimensions while concurrently preserving detailed feature information, as delineated in Eq. (2). Where the symbol "+" stands for branch aggregation. The enhancement feature $F_{EN}$ fuses the multiscale feature $F_{MPPM}$ and its cubic convolutional features by residual concatenation. And by applying $1 \times 1$ convolution and pooling operation on the multi-branch enhancement features, the channel dimension is unified, which in turn enhances the feature representation of defective sites.

The EM is instrumental in broadening the receptive field, reducing overfitting, and optimizing the representational capacity of samples across various scales.

$$F_{EN\_i} = F_{MPPM} + C(D_i(C(F_{MPPM})))$$
$$F_{EPPM} = P(C(F_{PPM}, F_{EN\_1}, F_{EN\_2}, F_{EN\_3})). \tag{2}$$

## Slim-neck fusing GSConv and VoV-GSCSP

Extensive computation in casting surface defect detection results in prolonged training time, posing challenges for practical implementation. The DES-YOLO adoption of a slim convolutional architecture known as GSConv (*Li et al., 2022*) is proposed to mitigate this. GSConv combines multiple convolutional operations, including standard convolution (Conv), depth-wise separable convolution (DSConv), and a Shuffle mechanism, as illustrated in Fig. 5A. This architecture reorganizes the input feature channels using Conv to facilitate the computation of features in multiple groups. It replaces the standard Conv of each group with DSConv and enhances inter-group feature integration by shuffling channel features. With a convolution kernel dimension of $K1 \times K2$, input feature map channels of $Ci$, output feature map channels of $Co$, and an output image dimension of $Wo \times Ho$. Traditional convolution has a single computation of $K_1 \times K_2 \times C_i \times C_o \times W_o \times H_o$, while DSConv convolution is divided into two parts: channel-by-channel convolution and point-by-point convolution. Channel-by-channel convolution uses only one convolution kernel per channel and has $K_1 \times K_2 \times C_i \times W_o \times H_o$ parameters, while point-by-point convolution uses $1 \times 1$ standard convolution of features and has $C_i \times C_o \times W_o \times H_o$ parameters. The computational load of GSConv is significantly reduced in comparison to

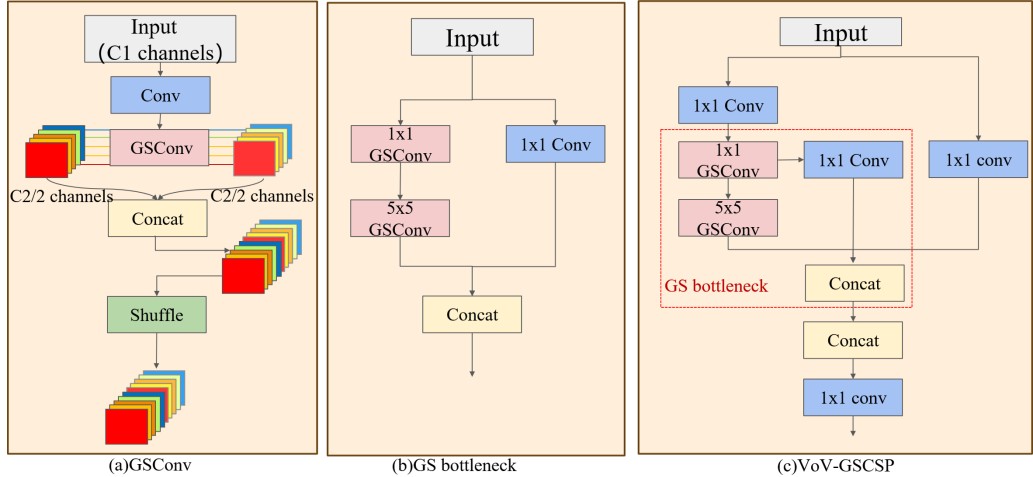

**Figure 5** **Slim-neck structure.** (A) The GSConv structure, (B) the GS bottleneck that comprises VoV-GSCSP, (C) the VoV-GSCSP module, which uses the residual idea. The combination of these three modules creates the slim-neck structure.

traditional Conv, as demonstrated in Eq. (3).

$$\frac{P_{DSConv}}{P_{Conv}} = \frac{K_1 \times K_2 \times C_i \times W_o \times H_o + C_i \times C_o \times W_o \times H_o}{K_1 \times K_2 \times C_i \times C_o \times W_o \times H_o} = \frac{1}{C_o} + \frac{1}{K_1 \times K_2} \qquad (3)$$

The VoV-GSCSP module, inspired by the ELAN architecture in YOLOv7, employs an unilayered aggregation strategy, as depicted in Figs. 5B and 5C. The VoV-GSCSP module uses dual GSConv to replace the traditional convolution operation. It uses the residual structure to sequentially splice shallow features with strong multi-scale positioning capabilities and deep features with strong multi-scale semantic expression capabilities into features that fuse multi-layer information. Vectors avoid the problems of information loss and gradient disappearance and reduce the complexity of calculation and network structure.

## Casting surface defect dataset -CSD DET

According to the Chinese national standard GB/T5611, casting surface defects are primarily classified into three categories: hole defects, incomplete defects, and surface defects, with each category further divided into seven subcategories. Hole defects arise from bubbles formed during casting when gas fails to escape during solidification. These are subdivided into blowholes, shrinkage cavities, and slag holes, distinguished by their visual characteristics. Insufficient metal liquid pouring or mechanical collision falls under incomplete defects. Surface defects, referring to general appearance issues in castings, are typically categorized into hollows and scratches, each resulting from varied manufacturing anomalies.

This study collected images of surface hole defects in textile machine parts, furnace castings from a hardware foundry in Xuancheng City, Anhui Province, and automobile cylinder block castings produced by an automobile industry company in Wuhu City.

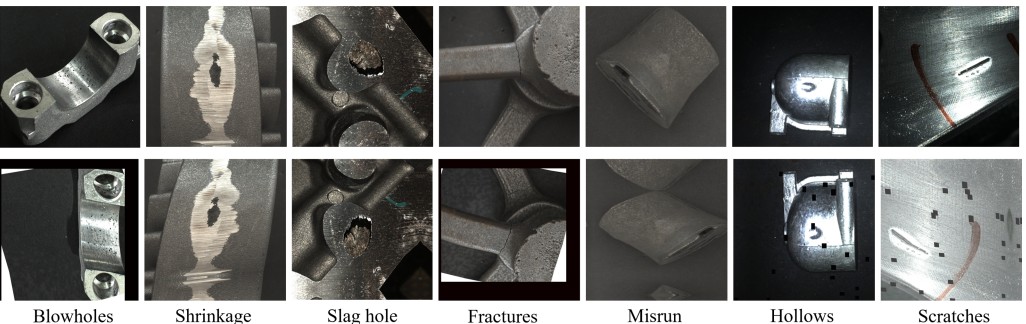

| Blowholes | Shrinkage | Slag hole | Fractures | Misrun | Hollows | Scratches |

**Figure 6  Comparison of surface defect types and data enhancement results of CSD-DET datasets.** The initial row shows the original dataset; the second row illustrates the CSD-DET dataset.

A Hikvision industrial camera was used to capture these images, allowing for the simulation of diverse production environments through adjustments to the camera aperture and background images. The camera features a maximum frame rate of 44.7 fps, an eight mm focal length lens, and an aperture range of F2.4 to F16. Images were captured at a resolution of 1,280 pixels × 1,280 pixels, with a shooting interval of 0.2 s. During data acquisition, special consideration was given to the defect samples' scale, angle, and viewpoint diversity to adapt to variations in target defects under different distances and sizes, aiming to improve the model's accuracy in defect detection and recognition. After a rigorous filtering process, 5,390 images of casting surface defects were obtained, each containing at least one defect—analysis of the collected images with 15% featuring multiple defects of varying sizes and uneven distribution. Multiple rounds of manual review were conducted to address the challenge of accurately labeling defect locations and categories in the dataset to ensure labeling accuracy and consistency. Additionally, the reliability of the labeling results was verified through random sampling and repeated labeling of selected data.

An analysis of category distribution within the dataset was conducted to address the imbalance in data samples caused by the low occurrence rates of shrinkage and hollow defects, and the sample numbers were balanced using random coefficients data enhancement techniques. These augmentation methods, including random luminance contrast adjustment, grid distortion, and affine transformations, were utilized to reduce data imbalance, as illustrated in Fig. 6. After applying these data enhancement techniques, the casting surface defect dataset, now encompassing 6,000 images with 8,854 defect markers, was designated as CSD-DET, as detailed in Table 1. Domain experts in casting were then engaged to evaluate the practical utility of the CSD-DET dataset for detecting surface defects in castings.

## RESULTS

### Loss functions

During the training of the DES-YOLO model, defect prediction frames are continuously generated. This study leverages the comparison between these prediction frames and

**Table 1  Dataset of casting surface defects.**

| Category | Original image/images | CSD-DET/ images | Label | Number of labels/instances |
|---|---|---|---|---|
| blowholes | 401 | 408 | Bh | 1,884 |
| shrinkage | 901 | 1,053 | Sh | 1,195 |
| slag hole | 842 | 915 | Sl | 1,516 |
| fractures | 850 | 903 | Fr | 1,348 |
| misrun | 1,002 | 1,009 | Mr | 1,250 |
| hollows | 634 | 1,002 | Ho | 1,101 |
| scratches | 660 | 710 | Sc | 1,260 |

ground truth (GT) frames to evaluate the model's ability to detect and accurately predict real defects. To achieve this, the SIOU loss function is utilized to optimize model parameters and enhance prediction accuracy by minimizing the loss. The SIOU loss function is employed, incorporating components such as confidence loss, coordinate regression loss, and target confidence loss, as detailed in Eq. (4). Here, $Loss_{classification}$ denotes the category confidence loss, which is used to determine whether the anchor frame is correctly classified. $Loss_{localization}$ denotes the coordinate regression loss, which is used to measure the error between the prediction frame and the real frame. $Loss_{confidence}$ denotes the target confidence loss, which evaluates whether the prediction frame contains the target or not. $\lambda$ is used as a super parameter to balance the effect of different loss functions on the backpropagation of model parameters by weighting coefficients,Throughout the model optimization process, with manual adjustments made multiple times and set to $\lambda 1 = 0.1$, $\lambda 2 = 0.005$, and $\lambda 3 = 0.1$ to meet the performance optimization requirements. The calculation of target and class confidence losses employs Binary Cross Entropy with Logits Loss (BCE With Logits Loss) combined with the sigmoid function.

$$Loss_{all} = \lambda_1 Loss_{classification} + \lambda_2 Loss_{localization} + \lambda_3 Loss_{confidence} \tag{4}$$

The YOLO model typically employs the Complete Intersection over Union (CIoU) as the coordinate regression loss function. This function facilitates prediction and refinement by evaluating distances and overlap areas between the predicted bounding boxes and the ground truth (GT) boxes. While the CIoU loss function aids in model optimization, it lacks consideration of the directional relationship between predicted boxes and GT boxes, resulting in the slow convergence speed of the model.

The DES-YOLO study employs the SIoU (*Zhora, 2022*) loss function, incorporating an area proportionality penalty term and a vector angle between projective regression. It comprises angle cost, distance cost, shape cost, and IoU cost, as depicted in Eq. (5). Where $\Delta$ represents the angular and distance loss coefficient and $\Omega$ represents the shape loss coefficient, $IoU$ represents the shape loss coefficient.

$$Loss_{SIoU} = 1 - IoU + \frac{\Delta + \Omega}{2} \tag{5}$$

To ensure smooth and controllable model convergence, an exponential decay function is employed to calculate the shape distance loss between predicted and actual frames,

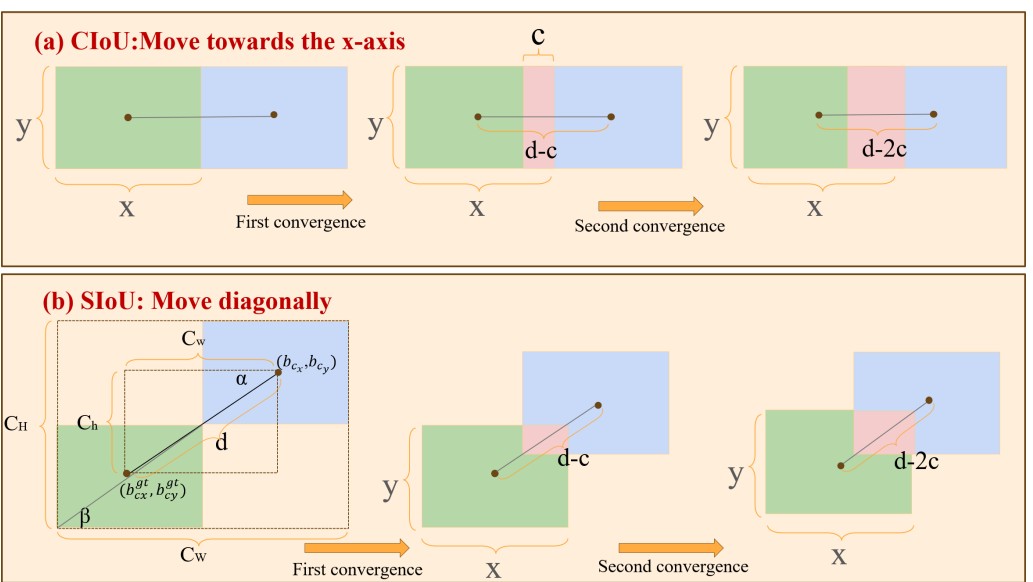

**Figure 7 Comparison of convergence between CIoU and SIoU.** (A) The convergence of CIoU moving along the $x$-axis; (B) the convergence of SIoU moving along the diagonal. The green and blue areas represent the ground-true box and predicted box, and the red area represents the overlap of the two boxes. The figure assumes that the prediction box is the same size as the GT box.

incorporating angular loss coefficients. The distance loss function $\Delta$ is associated with the minimum outer rectangle of the two bounding boxes, as defined in Eq. (6). Where $\Lambda$ denotes the angular loss coefficient, this coefficient is calculated by determining the minimum angles $\alpha$ and $\beta$ between the connecting line from the center point of the predicted bounding box to the actual bounding box and the $x$ and $y$ axes. The angular loss coefficient guides the predicted bounding box towards the direction of the axis of the nearest target box when the center point is not aligned with any of these axes ($\Lambda \neq 2$). The $\Delta$ loss is constrained to 0 when the center point of the predicted frame aligns with either the $x$-axis or $y$-axis, specifically when ($\Lambda = 2$). ($b_{cx}^{gt}, b_{cy}^{gt}$) represents the center coordinates of the ground truth (GT) box, and ($b_{c_x}, b_{c_y}$) represents the center coordinates of the predicted box. Additionally, $c_w$ and $c_h$ indicate the width and height of the minimum bounding rectangles for both the GT and predicted boxes, as illustrated in Fig. 7B. Equation $\Lambda$ Through the improved distance calculation formula, the angle $\alpha$ between the real frame and the center point of the prediction frame is determined, and the angle loss value is estimated using the multiplier angle formula.

$$\Lambda = 1 - 2 \times \sin 2(\arcsin(\frac{\max(b_{cy}^{gt}, b_{cy}) - \min(b_{cx}^{gt}, b_{cx})}{\sqrt{(b_{cy}^{gt} - b_{cy})^2 + (b_{cx}^{gt} - b_{cx})^2}}) - \frac{\Pi}{4})$$

$$\Delta = \sum_{t=x,y}(1 - e^{(\Lambda-2)\rho_t}) = 2 - e^{(\Lambda-2)(\frac{b_{cx}^{gt}-b_{cx}}{c_w})^2} - e^{(\Lambda-2)(\frac{b_{cy}^{gt}-b_{cy}}{c_h})^2}$$

(6)

In the shape loss function, the value of $\theta$ determines the unique Shape cost for each data point, as indicated in Eq. (7). The value of $\theta$ affects the gradient update step for the model's prediction box shape loss, and setting $\theta$ to 3 accelerates model training during the experimental manual parameter tuning process. The shape loss coefficients accumulate prediction errors across coordinates using an exponential decay model.

$$\Omega = \sum_{t=w,h} (1 - e^{-w_t})^\theta. \tag{7}$$

The width and height of the predicted and ground truth boxes are denoted by $x$ and $y$, respectively, with the distance between their center points represented as $d$ and the convergence speed as $c$. As depicted in Fig. 7A, the prediction frame shifts in the $X$-axis direction, progressively nearing the GT frame. The overlap of CIoU after the first convergence is denoted as $dc$, and after the second convergence, it becomes $2dc$. In contrast to CIoU's linear movement along the $x$-axis, SIoU steers the prediction frame toward the line connecting the center points of the predicted and ground truth frames. During this process, SIoU dynamically modifies the bounding box's dimensions to align with those of the ground truth box. Figure 7B illustrates the overlap degree between the two boxes after the two convergences.

The overlap degree for SIoU after the initial convergence is calculated as $\frac{xyc^2}{x^2+y^2}$, and it increases to $\frac{4xyc^2}{x^2+y^2}$ after the second convergence. Compared to CIoU, the overlap degree after movements guided by SIoU is significantly higher. This efficiency in diagonally directing the predicted bounding box's center point toward the real bounding box's center point facilitates faster model convergence during optimization.

## Experiment environment and evaluation index
### Experiment environment
The DES-YOLO model proposed in this research was trained and evaluated under controlled laboratory conditions. The experimental setup included hardware consisting of an Intel® Core™ i9-13900KS CPU at 3.20 GHz, an NVIDIA GeForce RTX 4090 GPU with 24 GB of memory, and software configuration conducted on Python 3.9 and PyTorch 2.0.0using the Anaconda3 deep learning development toolkit.

For consistency across tests, all images were standardized to a resolution of 1,280 pixels × 1,280 pixels. The CSD-DET dataset was divided into training, testing, and validation sets in an 8:1:1 ratio. The training process utilized stochastic gradient descent (SGD) as the optimizer, with a weight decay parameter set to 0.0005. The initial warm-up momentum was established at 0.8, the learning rate was initiated at 0.01, and the training was conducted over 300 epochs with a batch size of 8.

### Evaluation index
This study employs evaluation metrics such as precision (P), recall (R), F1 score, target detection accuracy, and mean average precision (mAP) to assess the effectiveness of casting surface defect detection. Precision represents the proportion of correct predictions relative to the total number of predictions, while recall measures the proportion of correct predictions relative to the actual number of predicted objects. Target detection accuracy

indicates the percentage of correctly identified defect samples relative to the total sample count. The F1 score measures the model's overall performance by precision and recall. mAP calculates the areas' average under the precision–recall (P-R) curve across all categories, offering a comprehensive assessment of the model's overall accuracy. The IoU threshold for calculating mAP is set to 0.5 and 0.5:0.95, as shown in Eq. (12).

$$Precision = \frac{TP}{TP+FP}. \tag{8}$$

$$Precall = \frac{TP}{TP+FN}. \tag{9}$$

$$Accuracy = \frac{TP+TN}{TP+FN+TN+FP}. \tag{10}$$

$$F1-score = 2 \times \frac{Precision \times Recall}{Precision+Recall}. \tag{11}$$

$$mAP = \frac{1}{n}\sum_{i=1}^{n}\int_{0}^{1}P_i(R_i)dR_i. \tag{12}$$

To evaluate the real-time performance of the casting surface defect detection model, the Infer Time metric is utilized to gauge the inference speed, which reflects the rate at which the model processes images per second. A lower Infer Time value indicates a faster processing speed, as detailed in Eq. (13).

$$Infer\_Time = \frac{the\_total\_number\_of\_images}{Total\_model\_check\_time}. \tag{13}$$

## Ablation experiments

This section details ablation experiments conducted to evaluate the effectiveness of individual modules within the DES-YOLO model for casting surface defect detection. The results are presented in Table 2, where the "✓"symbol denotes the inclusion of specific modules in the experiment. F1-score as an aggregate function of precision and recall. In the process of gradually introducing each module in the ablation experiment, the F1-score increased from 88.1 to 89.0, which proves that each module in the DES-YOLO model minimizes background misrecognition and enhances the accuracy of the correct identification of defective targets. The implementation of the DSC-Darknet referred to as A1, shows a detection accuracy that surpasses YOLOv7-tiny by 2.3% and 1.5%, underscoring the enhanced feature extraction capabilities of the DSC-Darknet backbone network.

Additionally, the inclusion of the GAM Attention and EPPM modules significantly boosts detection accuracy. While the slim-neck module enhances accuracy at the IoU threshold 0.5,

**Table 2  Ablation experiments.**

| Module | DSC-Darknet | GAM attention | EPPM | Slim-neck | SIoU | mAP@.5/% | mAP@.5:.95/% |
|---|---|---|---|---|---|---|---|
| YOLOv7 | | | | | | 87.3 | 65.4 |
| Algorithm1 | ✓ | | | | | 89.6 | 66.9 |
| Algorithm2 | ✓ | ✓ | | | | 90.5 | 67.6 |
| Algorithm3 | ✓ | ✓ | ✓ | | | 91.2 | 68.2 |
| Algorithm4 | ✓ | ✓ | ✓ | ✓ | | 92.4 | 67.9 |
| DES-YOLO | ✓ | ✓ | ✓ | ✓ | ✓ | 92.6 | 68.8 |

a slight reduction is noted at the 0.5:0.95 IoU threshold. This decrease is attributed to the slim-neck's shallow network depth, which, while reducing the computational load, aligns with practical application demands. Integrating the SIoU metric into the DES-YOLO model results in a mAP at 0.5 of up to 92.6% and a mAP at 0.5:0.95 of up to 68.8%, confirming the method's efficacy in identifying defects on casting surfaces.

While an attention mechanism is not imperative for target detection, experiments reveal that its application significantly enhances the model's ability to detect small target features, achieving accurate surface defect detection in inspection castings. Comparative analyses with mainstream mechanisms, such as ECA and NAM Attention, demonstrate the superior detection performance of GAM Attention, even without a substantial increase in parameter numbers. Although SK Attention exhibits the highest detection accuracy, its extensive requirement for calculating location information renders it less suitable for lightweight applications within factory settings. Notably, in the GAM attention experiment, accuracy rates exceed recall rates, a phenomenon that may be influenced by noise in the casting defect samples. This observation is corroborated by Fig. 8, which confirms that GAM attention is the most suitable for the DES-YOLO model.

The primary objective of the casting surface defect detection task is to effectively differentiate between background elements and characteristic features within the castings. A saliency heatmap is employed to demonstrate the detection model's focus on defects visually. This heatmap uses warmer image tones to represent higher model attention, indicating a greater probability that the highlighted region harbors defective features. Figure 9 illustrates how the model's attention to defective surface areas becomes more pronounced throughout the iterative training process.

In order to verify that the DES-YOLO model can effectively detect casting surface defects under different environments and lighting conditions, this study deployed the model in casting production line conveyor belts, casting storage warehouses, and other production environments. The test results are shown in Fig. 10, which proves that the DES-YOLO model has high generalization ability in different environments.

## Comparative experiments

To evaluate the improvement of the DES-YOLO model, this study presents a comprehensive comparison with the mainstream model, as summarized in Table 3. Compared to Faster R-CNN and RetinaNet, the YOLO series algorithms demonstrate significantly fewer

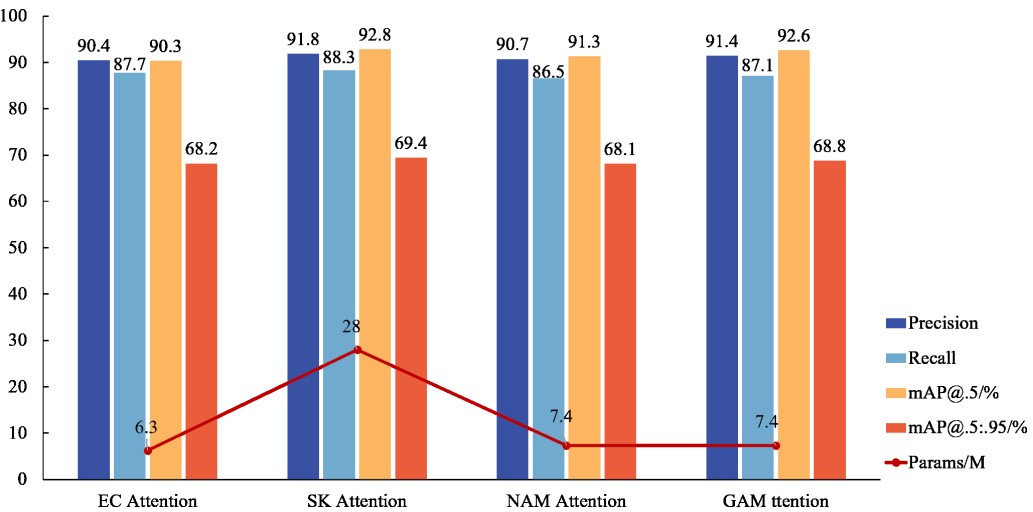

**Figure 8** Attention mechanism comparison.

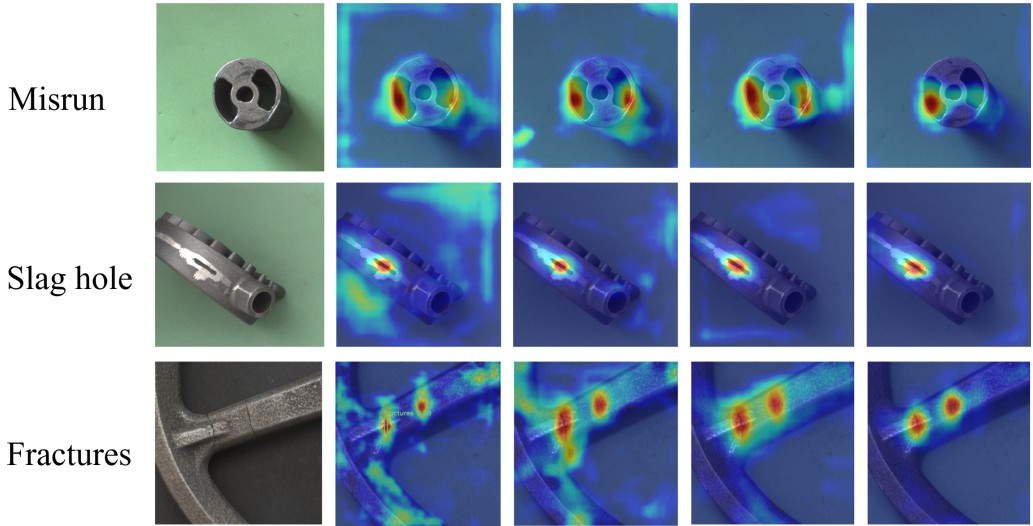

**Figure 9** DES-YOLO saliency heatmap. Rows of misrun, slag hole, and fracture defects, respectively. Each row in the figure illustrates the gradual integration of DSC-Darknet, MPPM module, slim-neck, and SIoU bounding box loss functions for detecting the same kind of faults.

parameters. Despite its higher parameter count, Faster R-CNN, a two-stage algorithm, attains the highest detection accuracy for scratch category defects at 85.1%. This superior performance is attributed to its Pyramid Region Network (PRN) characteristics and two-stage processing, effectively detecting minor target defects. In contrast, YOLOv3 demonstrates the lowest detection accuracy, with mAP's IoU threshold at 0.5 of only 84.3% and mAP's IoU threshold at 0.5:0.95 of 56.6%. The DES-YOLO model achieves a mAP's IoU threshold of 92.6%, reflecting a 5.3% improvement over the YOLOv7 model's 87.3% accuracy, primarily due to the utilization of DSC-Darknet and the EPPM model. The

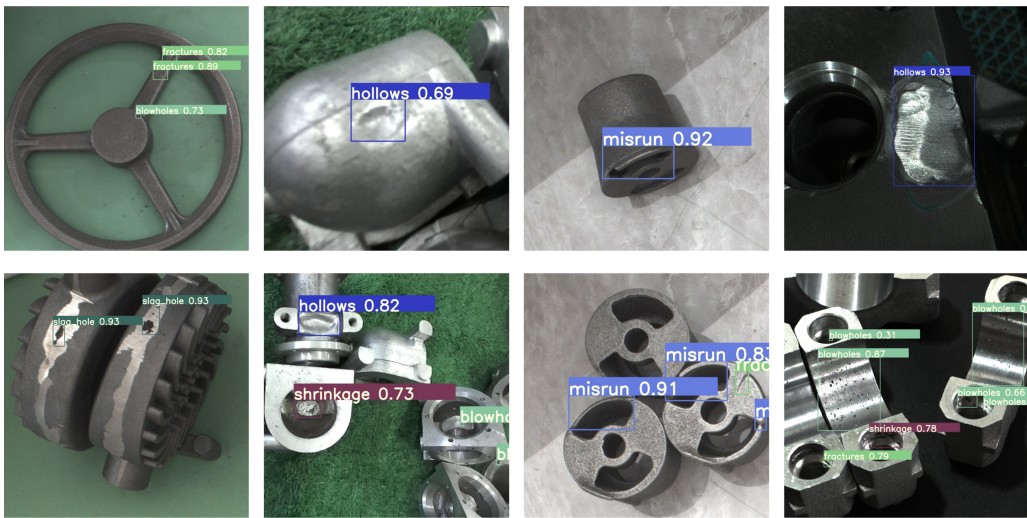

**Figure 10  DES-YOLO detection results in multiple scenarios.**

**Table 3  Compares the experimental accuracy.**

| Algorithm | Bl/% | Sh/% | Sl/% | Fr/% | Mr/% | Ho/% | Sc/% | mAP@.5/% | mAP@.5:.95/% |
|---|---|---|---|---|---|---|---|---|---|
| Faster R-CNN | 74.1 | 88.3 | 93.3 | 89.7 | 90 | 92.8 | 85.1 | 87.9 | 60.1 |
| Retina-net | 53.9 | 88.0 | 94.4 | 87.3 | 87.5 | 94.4 | 80.6 | 83.7 | 57.5 |
| YOLOv3 | 72 | 93.9 | 95.4 | 94.6 | 90.4 | 99 | 44.3 | 84.3 | 56.6 |
| YOLOv5 | 76.6 | 96.4 | 97.3 | 95.3 | 92.3 | 99.4 | 73.6 | 90.1 | 63.5 |
| YOLOR | 74.8 | 95.7 | 97.4 | 94.6 | 91.4 | 98.4 | 64.2 | 88.1 | 63.2 |
| PPYOLOE | 73.5 | 95.1 | 97.9 | 95.7 | 92.8 | 99.3 | 72.4 | 89.5 | 62.1 |
| YOLOX | 79.4 | 95.3 | 95.5 | 93 | 90.3 | 98.6 | 66.3 | 89.9 | 62.4 |
| YOLOv7 | 74.8 | 99.4 | 97.5 | 93.8 | 94.4 | 99.2 | 66.8 | 87.3 | 65.4 |
| DES-YOLO | 85.3 | 94.0 | 98.8 | 98 | 95 | 98.6 | 72.9 | 92.6 | 68.8 |

DES-YOLO model excels in detecting four types of defects: blowholes, slag holes, fractures, and misruns, with visualization results presented in Fig. 11.

To assess the DES-YOLO model's suitability for industrial real-time detection, its inference speed and computational complexity were analyzed and compared with other models, as detailed in Table 4. Faster R-CNN and RetinaNet lag behind the YOLO series regarding inference speed and computational complexity, posing challenges for industrial real-time detection. Although YOLOv3-tiny offers lower computational complexity, it does not achieve the inference speed or detection accuracy of the models discussed in this study.

In comparison, YOLOv5-s, YOLOR-p6, PPYOLOE-s, and YOLOX-s all demonstrate slower inference speeds, with their GFLOPs exceeding those of our model by 20.4%, 1.5%, 27.0%, and 95.6%, respectively. Notably, the DES-YOLO model achieves an inference speed of 3.9 ms, 0.5 ms faster than YOLOv7-tiny, while maintaining a computational load of only

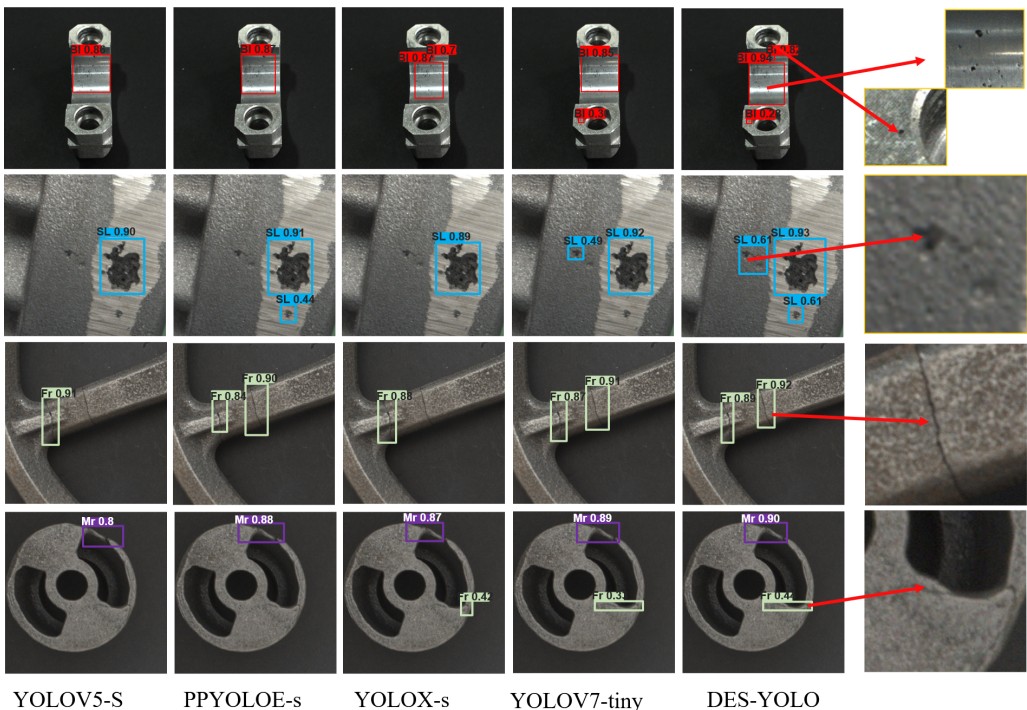

| YOLOV5-S | PPYOLOE-s | YOLOX-s | YOLOV7-tiny | DES-YOLO |

**Figure 11** **Comparison of the surface defect detection results of castings from the same model.** Each row shows a different defect type, such as blowholes, slag holes, fractures, and misruns. The final column of the figure includes magnified details of small, undetected target areas for each defect type.

**Table 4** **Computational complexity comparison experiment.**

| Model | Params/M | Infer time/ms | GFLOPs/$10^9$ |
| --- | --- | --- | --- |
| Faster R-CNN | 41.1 | 16.9 | 36.9 |
| RetinaNet | 36.2 | 35.3 | 30.8 |
| YOLOv3 | 8.7 | 5.1 | 13 |
| YOLOv5 | 12.3 | 13.7 | 16.5 |
| YOLOR | 8.4 | 14.5 | 13.9 |
| PPYOLOE | 36.9 | 10.6 | 17.4 |
| YOLOX | 8.06 | 15.2 | 26.8 |
| YOLOv7 | 6.3 | 4.4 | 13.2 |
| DES-YOLO | 7.4 | 3.9 | 13.7 |

13.7 GFLOPs. This efficiency satisfies the stringent requirements for real-time detection of casting surface defects. Analysis of the loss attenuation curves, as shown in Fig. 12, reveals that the DES-YOLO model's confidence loss relative to total loss declines more

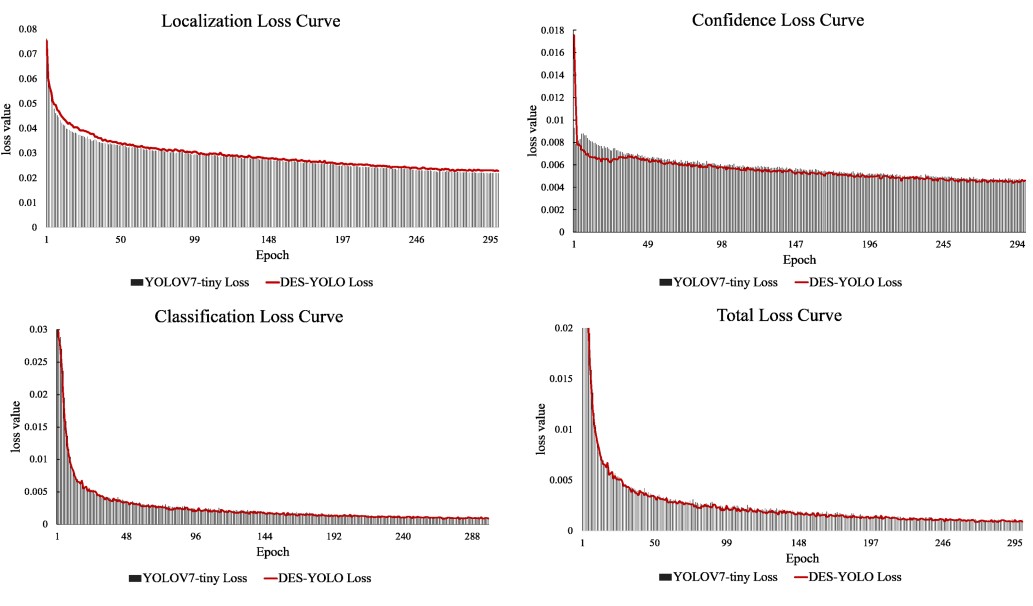

**Figure 12 Loss attenuation curve comparison.**

smoothly and rapidly than YOLOv7-tiny, indicating enhanced learning and generalization capabilities.

The analyses above further corroborate that our proposed DES-YOLO model exhibits superior robustness in the real-time detection of casting surface defects.

## CONCLUSIONS

The DES-YOLO model is designed for real-time detection of casting surface defects, effectively addressing challenges such as limited datasets, similarity among defective features, and the complexity of extracting features from small targets. The effectiveness of this improved method has been validated through experimental validations, with critical conclusions as follows:

1. DSC-Darknet backbone network: Engineered to enhance the stability of the DES-YOLO model during feature extraction, this network integrates the DSC layer to reduce feature computation demands, the DBC block to enrich gradient combinations, and the GAM Attention module to amplify spatial feature information, facilitating the extraction of features from small targets.

2. Enhanced pyramid pool module: The EPPM is integrated into the feature extraction network to expand the receptive field, enhancing the network's ability to detect target samples of varying sizes. The EPPM module significantly boosts the accuracy of small-casting surface defect detection.

3. Slim-neck: By integrating the GSConv and VoV-GSCSP modules, the Slim-neck configuration effectively simplifies the DES-YOLO model's structure and reduces the detection network's depth. This strategic reduction significantly lowers the computational

memory overhead, enhancing the model's efficiency. This module has led to a significant improvement in the applicability of the DES-YOLO model in industrial scenarios.

4. SIoU bounding box regression loss function: This function, which includes an area proportion penalty term and vector angle consideration, accentuates the logical relationship between predicted and actual bounding boxes, thereby enhancing the model's fitting stability.

The DES-YOLO model outperforms mainstream target detection algorithms, achieving mAP of 92.6% and 68.8% on the CSD-DET dataset for casting surface defect detection, with a single image detection time of 3.9 ms and a computational complexity of 13.7 GFLOPs. The experimental results affirm the efficacy of the DES-YOLO model in identifying surface defects in casting.

Future research will concentrate on the following areas to enhance the DES-YOLO model's practical applications and address existing challenges:

1. Practical application development: Efforts will be dedicated to exploring the practical needs associated with the DES-YOLO model in industrial settings. The research includes developing a comprehensive casting surface defect detection system that integrates casting conveyor belts, acquisition cameras, and sophisticated detection systems to streamline and automate the inspection process.

2. Lighting variation challenges: The research will also address the significant challenge of accurately extracting defective features in casting images that are affected by variations in industrial lighting. The research involves improving the model's robustness in different lighting conditions.

## ACKNOWLEDGEMENTS

The authors are grateful to all the Intelligent Mechanics and Robotics research team members of the School of Artificial Intelligence of Anhui University of Science and Technology for their advice and help in this study.

### Funding

This paper was supported by Natural Science Foundation of Anhui Province: 2208085ME128 and the National Natural Science Foundation of China, Grant/Award Number: 61873004. The funders had no role in study design, data collection and analysis, decision to publish, or preparation of the manuscript.

### Grant Disclosures

The following grant information was disclosed by the authors:
Natural Science Foundation of Anhui Province: 2208085ME128.
National Natural Science Foundation of China: 61873004.

### Competing Interests

The authors declare there are no competing interests.

## Author Contributions

- Chengjun Wang conceived and designed the experiments, analyzed the data, authored or reviewed drafts of the article, and approved the final draft.
- Jiaqi Hu conceived and designed the experiments, performed the experiments, analyzed the data, performed the computation work, authored or reviewed drafts of the article, and approved the final draft.
- Chaoyu Yang performed the experiments, performed the computation work, prepared figures and/or tables, and approved the final draft.
- Peng Hu analyzed the data, prepared figures and/or tables, and approved the final draft.

## Data Availability

The raw data and code are available in the Supplemental Files.

## Supplemental Information

Supplemental information for this article can be found online at http://dx.doi.org/10.7717/peerj-cs.2224#supplemental-information.

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
