# Peer review of "DES-YOLO: a novel model for real-time detection of casting surface defects"

_PeerJ Computer Science, doi:10.7717/peerj-cs.2224_

## Round 0.1 · original submission · Major Revisions

Consider the following reference:
1. https://doi.org/10.1016/j.ifacol.2018.09.412

**Language Note:** PeerJ staff have identified that the English language needs to be improved. When you prepare your next revision, please either (i) have a colleague who is proficient in English and familiar with the subject matter review your manuscript, or (ii) contact a professional editing service to review your manuscript. PeerJ can provide language editing services - you can contact us at [email protected] for pricing (be sure to provide your manuscript number and title). – PeerJ Staff

Reviewer 1 ·

Basic reporting

No comment

Experimental design

No comment

Validity of the findings

No comment

Additional comments

No comment

Cite this review as

Reviewer 2 ·

Basic reporting

No comment

Experimental design

No comment

Validity of the findings

No comment

Additional comments

1. This paper proposes a real-time detection method for casting surface defects named DES-YOLO using computer vision technology. In order to solve the problem of high defect feature similarity, a DSC-Darknet backbone network is introduced to optimize the defect feature extraction, and a GAM attention mechanism is introduced to increase attention to the defective target features in order to distinguish the semantic information between diûerent features.
2. Graphical represantitions for results can be given.
3. Computational comlexity of the proposed approach should be given to show applicability and usability of the proposed approach.
4. Additional results can be provided for different scenarios and/or real conditions.

Cite this review as

Reviewer 3 ·

Basic reporting

This paper presents a novel DES-YOLO object detection algorithm for casting, enhancing the precision of surface defect detection. It has practical potential for automating quality control in the foundry industry. The structure is reasonable and logical, and the design principle is clearly and moderately illustrated.

Experimental design

no comment

Validity of the findings

no comment

Additional comments

This paper presents a novel DES-YOLO object detection algorithm for casting, enhancing the precision of surface defect detection. It has practical potential for automating quality control in the foundry industry. The paper is well written and organized. The structure is reasonable and logical, and the design principle is clearly and moderately illustrated. However, there are some problems to be further improved as well:
1. Take care of the syntactic structure and supplement sentences with incomplete subject-verb-object elements, such as line 161 of the manuscript.
2. More relevant works on data enhancement should be introduced in Section I to support the authority of the argument.
3. The figure of the DES-YOLO algorithm structure should be presented in the text to clearly emphasize the method's innovation.
4. Organize the table (4) and clarify the unit of each item.
5. Provide a detailed explanation of the relevant variables for the equation (6).
6. Provide a detailed interpretation of Graph 3 and add the annotated description of the components in the graph.

Cite this review as

Reviewer 4 ·

Basic reporting

no comment

Experimental design

no comment

Validity of the findings

no comment

Additional comments

Rigorous research was conducted in accordance with high technical and ethical standards. The research question is well defined, relevant and meaningful. It is stated how the research fills an identified knowledge gap. The results are well articulated, related to the original research question and limited to supporting conclusions.

Cite this review as

---

## Round 0.2 · Minor Revisions

Address reviewers' comments and improve the quality of presentation and coherence in this work.

**Language Note:** The review process has identified that the English language must be improved. PeerJ can provide language editing services - please contact us at [email protected] for pricing (be sure to provide your manuscript number and title). Alternatively, you should make your own arrangements to improve the language quality and provide details in your response letter. – PeerJ Staff

Reviewer 3 ·

Basic reporting

Thanks so much for the authors' response and revision. I think this paper can be accepted in this form.

Experimental design

Thanks so much for the authors' response and revision. I think this paper can be accepted in this form.

Validity of the findings

Thanks so much for the authors' response and revision. I think this paper can be accepted in this form.

Cite this review as

Reviewer 5 ·

Basic reporting

Although the language improvements are evident, some sentences still appear to be slightly unclear. It is recommended to have a professional language editing service review the manuscript to refine the language for clarity and fluency.

Experimental design

The methods section is detailed and should allow replication of the study. The use of the DSC-Darknet backbone, GAM attention mechanism, and other technical enhancements are well-explained.

Validity of the findings

1. It would be helpful to include axis labels, legends, or explanatory notes in some of the figures where they are missing or unclear.
2. More details about the CSD-DET dataset would be useful. Information such as how the dataset was compiled, any biases in the data, and limitations should be explicitly mentioned.

Additional comments

The provided description of comparative analysis between the DES-YOLO model and other models like Faster R-CNN, RetinaNet, and YOLO series seems to be informative in terms of inference speed and computational complexity. However, as the authors claim their method is highly robust compared to the other models, it would be useful to discuss the robustness of DES-YOLO in various operational conditions, like varying lighting or angles, and compare it with the other models.

Cite this review as

---

## Round 0.3 · Major Revisions

Equation 1:
Assumption of Linearity in Feature Aggregation: The equation assumes a linear combination of pooled features is sufficient to capture the complex spatial hierarchies in image data. This may oversimplify the interactions between features at different scales, potentially overlooking complex patterns that a non-linear aggregation method might capture.

Fixed Pooling Operations: The use of fixed pooling operations (average pooling) may not be optimal for all types of features. Adaptive pooling methods that can adjust based on the feature characteristics might offer more flexibility and effectiveness in capturing relevant information.

Bilinear Upsampling: While bilinear upsampling is popular for its simplicity and effectiveness in certain scenarios, it may introduce artifacts or fail to capture high-frequency details. More sophisticated upsampling methods, such as learned transposed convolutions or pixel shuffle operations, could provide better quality feature maps.

Equation 2:
Convolution with Fixed Expansion Rates: The use of fixed expansion rates in the convolution branches may not optimally capture the variability in feature size and shape present in casting surface defects. A dynamic or learnable expansion rate might improve the model's ability to adapt to different defect characteristics.

Weight Sharing Across Branches: While weight sharing reduces the parameter count, it also constrains the model's capacity to learn diverse feature representations. This could limit the model's expressiveness and adaptability to complex defect patterns.

Statistical Analysis :
Generalization of Results: The experiments are conducted on a specific dataset, which may limit the generalizability of the findings. This proposed paper compares with different algorithm and models but It's crucial to evaluate the model's performance across diverse datasets to ensure its robustness and applicability in real-world scenarios.

Loss Function Design: The design of the loss function, especially the use of Binary Cross Entropy with Logits Loss combined with the sigmoid function, might not be the most effective approach for all types of defects. Exploring alternative loss functions, such as focal loss or dice coefficient, could provide benefits in handling class imbalance and enhancing model sensitivity to rare defects.

Evaluation Metrics: While precision, recall, and mAP are standard metrics, the study could benefit from incorporating additional metrics such as F1 score, IoU, and object detection accuracy to provide a more comprehensive assessment of model performance.

Computational Efficiency: Despite efforts to reduce computational demand, the complexity and depth of the proposed model might still pose challenges for deployment in real-time industrial environments. Further optimization, perhaps through network pruning or quantization, could enhance efficiency without significantly compromising accuracy.

The paper writing quality is quite weak -- it seems that more assistance is gained from the LLM models - kindly use English editing service or professionals to edit writing/formatting and placement of equations properly in proper format.

**Language Note:** The review process has identified that the English language must be improved. PeerJ can provide language editing services - please contact us at [email protected] for pricing (be sure to provide your manuscript number and title). Alternatively, you should make your own arrangements to improve the language quality and provide details in your response letter. – PeerJ Staff

---

## Round 0.4 · Minor Revisions

Make all equations in the paper in a more professional way - and give its meaning and significance and context of how it is used. Make the writing more professional and make it justified so that it looks clear to the reader.

Reviewer 3 ·

Basic reporting

The paper is well written and organized

Experimental design

The experiment is well designed and presented

Validity of the findings

The work is of signifcant engineering value

Additional comments

The paper can be accepted in this form

Cite this review as

Reviewer 5 ·

Basic reporting

The manuscript is written in clear, professional English suitable for an international audience.
It provides a thorough background and relevant literature references, establishing a strong context for the study.

Experimental design

The experimental design is robust, well-detailed, and adheres to high technical and ethical standards. The detailed descriptions ensure the replicability of the study.
Innovations such as the DSC-Darknet backbone, GAM attention mechanism, and SIoU bounding box regression are clearly explained and justified.

Validity of the findings

The manuscript effectively addresses initial concerns about figure annotations and dataset details. Figures are now well-labeled, and the dataset is thoroughly described, including its compilation, potential biases, and limitations.
The findings are robust, statistically sound, and the conclusions are well supported by the results

Additional comments

The authors have addressed all major concerns raised during the initial review. The improvements made to the manuscript significantly enhance its quality and contribute meaningfully to the literature on surface defect detection using deep learning.
While the current manuscript is strong, continuous updates and checks on dataset biases should be encouraged as part of future work to ensure the model's applicability and robustness in industrial settings.
Consider exploring the integration of the model into a real-time industrial system for practical validation beyond controlled experiments.

Cite this review as

---

## Round 0.5 · Minor Revisions

The writing is not very professional especially the way the equations are outlined. kindly make if more appealing for the readers. Also check the formula for Precall -- this looks similar to Recall or is there a typo? . Also loss function in Equation 4 and 5 are not well defined and incorrect... check this out. In the shape loss function, the value of θ determines the unique Shape cost for each data point.

Why this study's hyperparameter θ is set to 3. Equation (7) is not explained clearly. The shape loss coefficients accumulate prediction errors across coordinates using an exponential decay model.

page 9, line 289 " Equation ^ Through the improved distance .... " check this line. What is delta in Eq 6 (also explain Eq 6). What is Omega in Eq 7 (explain it). Make sure there is smooth flow from one section to another for clear coherence for the reader.

---

## Round 0.6 · accepted · Accept

The overall presentation / writing style could have been improved. But I am Accepting the manuscript at this time.